# The Genus *Kalanchoe* (Crassulaceae) in Ecuador: From Gardens to the Wild

**DOI:** 10.3390/plants11131746

**Published:** 2022-06-30

**Authors:** Anahí Vargas, Ileana Herrera, Neus Nualart, Anne Guézou, Carlos Gómez-Bellver, Efraín Freire, Patricia Jaramillo Díaz, Jordi López-Pujol

**Affiliations:** 1Escuela de Ciencias Ambientales, Universidad Espíritu Santo (UEES), Samborondón 091650, Ecuador; anahivargas5@gmail.com; 2Instituto Nacional de Biodiversidad (INABIO), Quito 170501, Ecuador; efrain.freire@biodiversidad.gob.ec; 3Botanic Institute of Barcelona (IBB), CSIC-Ajuntament de Barcelona, Barcelona 08038, Catalonia, Spain; nnualart@ibb.csic.es (N.N.); carlosnuriabcn@gmail.com (C.G.-B.); 4Independent Researcher, Puerto Ayora 200350, Ecuador; anne.guezou8@gmail.com; 5Colecciones de Historia Natural, Estación Científica Charles Darwin, Galapagos Islands 200102, Ecuador; patricia.jaramillo@fcdarwin.org.ec; 6Facultad de Ciencias, Universidad de Málaga, 29071 Málaga, Spain

**Keywords:** invasive alien species, biological records, *Bryophyllum*, Galapagos Islands, protected areas

## Abstract

The genus *Kalanchoe*, mostly indigenous from Madagascar and Tropical Africa, is widely traded for ornamental value. In this study, we provided an updated list of wild and cultivated *Kalanchoe* taxa in Ecuador; we analyzed the temporal–spatial pattern of their records, and we categorized the invasion status for each taxon and its environment preferences. The records of any taxa belonging to this genus were compiled from an extensive search using various information sources. Our results confirmed the presence of 16 taxa of *Kalanchoe* in the country. Seven species and a hybrid were detected in the wild. *Kalanchoe densiflora*, *K. laxiflora*, *K. pinnata*, *K. tubiflora*, and *K.* ×*houghtonii* were categorized as invasive. We detected invasive records of some of these plants in protected areas. Almost all taxa had at least one record as cultivated, suggesting that the invasion pathway is ornamental trade. *Kalanchoe pinnata* individuals in the wild were recorded in the four biogeographic regions of Ecuador, which could be associated with the wide range of precipitations and temperatures in which the species may dwell. Our study highlights the importance of reducing the ornamental value and limiting the use of *Kalanchoe* taxa with invasive potential in horticulture and promoting, instead, the use of indigenous species.

## 1. Introduction

Alien species are those transported beyond their natural geographical distribution range by human activities [1,2]. Some of the alien species could become invasive alien species (IAS) and, thus, being of concern for their new environment. In the recipient ecosystems, IAS expand their distribution, maintain high-density populations, and can generate impacts on the ecosystem, economy, and/or human health [3,4]. Biological invasions are a function of propagule pressure, species-specific traits (e.g., high reproductive output), and the vulnerability of recipient ecosystems to biological invasions (e.g., frequency of disturbance) [5,6]. Today, the entire invasion process is favored by the growth in the number of species transported by humans combined with the constant increase in anthropic disturbances [3,7]. In fact, a high propagule pressure is generally associated with deliberate introductions rather than accidental ones [8].

The global trade of ornamental plants is one of the dominant pathways by which alien species have been introduced deliberately worldwide [8,9,10,11]. For example, half of the alien flora in Germany was introduced deliberately, with more than 50% of alien plants having arrived as ornamental ones [12]. In the Czech Republic, 53% of alien plant taxa were also introduced as ornamental ones [13]. A study on introduction pathways of alien species in Australia for the period of 1971–1995 showed that 65% of 290 newly established, non-native plants were introduced as ornamental plants for horticulture [14]. Furthermore, in the United States more than 50% of naturalized species were deliberately introduced [14]. Recent data from China suggest similar patterns: up to 56.5% of alien species were introduced for their ornamental value, and half of these have become naturalized [15].

The genus *Kalanchoe* (Crassulaceae) is well known for its ornamental value [16,17,18], and some taxa of the genus are also used for traditional medicine in various countries [19,20]. *Kalanchoe* comprises approximately 150 species, naturally mostly distributed in Madagascar and continental Africa, but also extending from Arabia to Southeast Asia [21]. Most taxa in the genus *Kalanchoe* are herbs with succulent leaves and clustered flowers. They are commonly used in gardening and by florists due to their easy cultivation, colorful flowers, adaptability to drought, vigorous clone growth, and long flowering time [20,22]. Many taxa of *Kalanchoe* reproduce both sexually and clonally, with the plantlets asexually produced being variously carried along the margins of the leaves or, typically post-anthesis, on the inflorescences. This asexual reproduction is so prolific that they sometimes escape into the wild [16,23,24].

There is a wide range of ethnomedical applications of the genus *Kalanchoe* that are related to their chemical composition. Indeed, in South America, some taxa are called *hojas milagrosas* (in English: “miraculous leaves”). The aerial parts and juice are used externally to treat inflammation, allergies, and different skin disorders [25,26,27]. Some *Kalanchoe* taxa are a source of bufadienolides, with cardioactive properties and anticancer activity (for a review, see [28]). A recent study reported that *K. blossfeldiana* ethanol extract is a potential candidate for cancer and bacterial infection treatment [29]. In contrast, another study shows evidence that *K. pinnata*, a widely used medicinal plant, may cause DNA damage and its use should be restricted [30]. *Kalanchoe* plants are also well known for causing episodes of poisoning of cattle and domestic pets [16,31,32,33].

The high ornamental value and frequent use in popular medicine of some taxa of *Kalanchoe* have undoubtedly increased their commercialization around the world, favoring their invasion success [19,23,34,35]. Several taxa of the genus *Kalanchoe* have been categorized as invasive or, at least, as taxa of concern for their new environment [19]. For example, in Australia, *K. tubiflora* is an invasive species and it was estimated to occupy more than 10,000 ha already four decades ago [36], while *K. daigremontiana* has been reported as invasive in the continental United States [37], Cuba [35], Hawaii [38], and Australia [19], among others. *Kalanchoe* ×*houghtonii*, an artificial hybrid between *K. daigremontiana* and *K. tubiflora*, is considered an emerging invader on a global scale, a great example of how the development of artificial hybrids in greenhouses can give rise to new biological invasions [39]. *Kalanchoe pinnata* is widely naturalized in the tropics, and its wide distribution as an alien species is associated with its use as a medicinal plant [19,40,41].

Ecuador, one of the megadiverse countries of the world, is not an exception of *Kalanchoe* expansion. In continental Ecuador, three *Kalanchoe* species (*K. blossfeldiana*, *K. daigremontiana,* and *K. pinnata*) and a hybrid (*K.* ×*houghtonii*) have been reported to date [42]. In the Galapagos Islands, seven species have been reported: *K. blossfeldiana*, *K. daigremontiana*, *K. eriophylla*, *K. fedtschenkoi*, *K. gastonis-bonnieri*, *K. pinnata,* and *K. tubiflora* [43,44,45]. *Kalanchoe pinnata* is especially problematic in the Galapagos Islands, where one of its main impacts is the formation of thick stands that displace the existing vegetation, reducing local biodiversity and also forming a dense carpet that inhibits the regeneration of indigenous species [46]. Records of various observers on the iNaturalist platform (available at https://www.inaturalist.org/observations?locale=es-ES&place_id=137386&taxon_id=59379; accessed on 1 March 2020–31 October 2021), however, suggest the presence of other *Kalanchoe* taxa in Ecuador. No previous study has examined the distribution and possible impacts of taxa of this alien genus in Ecuador.

The aims of the present study are the following: (1) to provide an updated and reliable list of wild and cultivated taxa of the genus *Kalanchoe* in Ecuador (including the Galapagos Islands), which is complemented with an identification key; (2) to analyze the temporal accumulation and spatial distribution of all *Kalanchoe* records (as a whole and sorted by taxa); (3) to explore the environmental preferences of *Kalanchoe* taxa occurring in the wild in Ecuador; and (4) to estimate the invasion status (i.e., casual, naturalized, or invasive) at the country level and, for the case of invasive taxa, to discuss their possible impacts. Finally, we examined the presence of *Kalanchoe* records within the Ecuadorian protected areas (PAs).

## 2. Results

### 2.1. Richness of Kalanchoe Taxa in Ecuador

We collected 967 verified records of *Kalanchoe* in Ecuador (Table 1). The Andean Region had the highest number of records (71%, *N* = 684), followed by the Insular Region (13%, *N* = 127), the Amazonian Region (11%, *N* = 105), and the Coastal Region (5%, *N* = 51). We confirmed the presence of 16 taxa of *Kalanchoe* in the country. *Kalanchoe laxiflora* had the highest number of records with 29% (*N* = 280), followed by *K. pinnata* with 26% (*N* = 250), and *K. densiflora* with 19% (*N* = 182) (Table 1). The Andean Region, again, had the highest number of *Kalanchoe* taxa (13 taxa; Table 1). Ten taxa were recorded for the Coastal Region and nine taxa were confirmed for both the Amazonian and Insular regions (Table 1). Up to five taxa of *Kalanchoe* were present in all four bioregions of Ecuador (*K. blossfeldiana*, *K. daigremontiana*, *K. gastonis-bonnieri*, *K. pinnata,* and *K.* ×*houghtonii*) (Table 1). In the Andean Region, the species most frequently recorded were *K. laxiflora* (41%, *N* = 277) and *K. densiflora* (27%, *N* = 180) (see Table 1). *Kalanchoe pinnata*, in contrast, was the species most frequently recorded in the Amazonian Region (87%, *N* = 91), the Insular Region (61%, *N* = 78), and the Coastal Region (37%, *N* = 19) (see Table 1).

### 2.2. Key to Identify the Taxa of Kalanchoe Introduced in Ecuador Detected in the Wild with Certainty

The botanical key for the taxonomic identification of the *Kalanchoe* taxa occurring in the wild in Ecuador is as follows (note: the term “bulbil” is used here to refer to pseudo-bulbils or incipient seedlings, although the word in the strict sense refers to underground or aerial buds that function as an organ of vegetative multiplication).
1.Leaves pinnately compound, at least in part………………………..………..*K. pinnata*2.Leaves always simple
2.1.Leaf blade suborbicular, ovate, or elliptic (length/width ratio between 1:1 and 2:1)
2.1.1.Plants 0.3–1(3) m. Green leaves without spots, finely crenate. Inflorescence of erect flowers. The only taxon of the group under study that does not produce bulbils on the leaves……………………………………..*K. densiflora*2.1.2.Plants up to 0.5 m. Green or somewhat greyish leaves, sometimes with scattered brown spots and a reddish line on the edge, which may have small auricles at the base, with a crenate margin with few lobes. Inflorescence of pendulous flowers…………………………………...…*K. laxiflora*2.2.Leaf blade narrowly ovate, oblong, subdeltoid, linear, or subcylindrical (ratio 3:1 or greater)
2.2.1.Plants with narrowly ovate, oblong, or subcylindrical leaves, crenate, with spots on the reverse or without spots, which form bulbils only at the apex
2.2.1.1.Narrowly ovate to oblong leaves
2.2.1.1.1.Green or bluish-green leaves generally without spots*…..K. mortagei*2.2.1.1.2.More or less light green leaves, with irregular spots………………………………………………...……..*K. gastonis-bonnieri*2.2.1.2.Subcylindrical leaves…………………………………………...*K. tubiflora*2.2.2.Plants with narrowly ovate, oblong, or subdeltoid leaves, toothed, with spots on the reverse, which form bulbils along almost the entire leaf margin
2.2.2.1.Leaves subdeltoid or triangular, the largest subpeltate with the base forming a very conspicuous fold perpendicular to the blade………………………………………………………...*K. daigremontiana*2.2.1.2.Leaves variable in shape and size, from narrowly lanceolate with a decurrent base towards the petiole to non-decurrent subdeltoid, which do not form a basal fold………………………………………..*K. ×houghtonii*

### 2.3. Temporal Accumulation and Distribution of Kalanchoe Records

Figure 1a shows the year and location of the oldest record for each taxon of *Kalanchoe*, either wild or cultivated. The oldest occurrences of five species (*K. carnea*, *K. eriophylla*, *K. gastonis-bonnieri*, *K. pinnata*, and *K. tubiflora*) were recorded in the Galapagos Islands (Figure 1a). The first record for the remaining taxa of *Kalanchoe* currently present in Ecuador came from the Andean Region, except for *K. daigremontiana* and *K*. ×*laetivirens* that were firstly reported in the Coastal Region (provinces of Los Rios and Manabí, respectively; Figure 1a). Although the first record of *Kalanchoe* in Ecuador was in the Galapagos Islands in 1905 and corresponded to *K. pinnata*, the other species were not observed until much later (starting with *K. crenata* and *K. fedtschenkoi*, reported in Azuay province in 1981; Figure 1a,b). A detailed list of the oldest records, both in the wild and cultivated, for each *Kalanchoe* taxon reported in Ecuador is provided in Appendix A.

Figure 2a,b shows the spatial and temporal accumulation of *Kalanchoe* records. The highest quantity of reports was in the Andean Region, particularly in the Pichincha and Imbabura provinces during the last five years (Figure 2a,b). With the relative exception of *K. pinnata* (whose number of occurrences increased quite steadily between 1939 and the late 2010s), the accumulated records for each taxon have grown exponentially since the late 2010s, with very abrupt growths for *K. densiflora*, *K. laxiflora*, and *K.* ×*houghtonii* (Figure 2b).

### 2.4. Kalanchoe Taxa Categorized

Almost all the taxa had at least one record as cultivated, with the exception of *K. thyrsiflora* and *K*. ×*laetivirens*, which had all their occurrences classified as uncertain (Figure 3a). Six species had all their records classified as cultivated or a mix of cultivated/uncertain (*K. blossfeldiana*, *K. carnea*, *K. crenata*, *K. eriophylla*, *K. fedtschenkoi*, and *K. tomentosa*) (Figure 3a). The remaining eight taxa (*K. daigremontiana*, *K. densiflora*, *K. gastonis-bonnieri*, *K. laxiflora*, *K. mortagei*, *K. pinnata*, *K. tubiflora*, and *K.* ×*houghtonii*) showed both wild and cultivated records (Figure 3a). By regions, the Andean was the one with most *Kalanchoe* taxa in the wild, with seven taxa, followed by the Amazonian and Insular regions, both with four taxa (Figure 3). *Kalanchoe pinnata* was the only species of the genus that showed wild occurrences in the four regions of Ecuador, whereas up to five taxa were cultivated along all bioregions: *K. blossfeldiana*, *K. daigremontiana*, *K. gastonis-bonnieri*, *K. pinnata*, and *K.* ×*houghtonii*. Some taxa, in contrast, were scarcely distributed, either as wild (*K. daigremontiana* and *K. gastonis-bonnieri*, present only in the Andean Region and in the Galapagos Islands, respectively) or cultivated (*K. carnea*, *K. eriophylla*, and *K. tubiflora* were cultivated only in the Galapagos Islands, while *K. crenata*, *K. densiflora*, and *K. thryrsiflora* did exclusively in the Andean Region) (Table 1 and Figure 3).

Regarding the eight taxa with at least one record in the wild, *K. mortagei* was the only one categorized as casual in Ecuador, since one of their two wild occurrences was casual and the other uncertain (Figure 4a). As two species (*K. daigremontiana* and *K. gastonis-bonnieri*) had at least one population fully naturalized (Figure 4a), they were classified as naturalized species at the national level. The remaining five taxa (*K. densiflora*, *K. laxiflora*, *K. pinnata*, *K. tubiflora*, and *K*. ×*houghtonii*) were regarded as invasive because they had at least one record behaving as such. However, only two of these five taxa (*K. laxiflora* and *K. pinnata*) had a sizeable proportion of their wild occurrences (about one third) as invasive (Figure 4a). With the exception of *K. pinnata*, no taxa had records categorized as naturalized or invasive in the Amazonian and Coastal regions; in contrast, the naturalized and invasive populations for several taxa were common in both the Andean and Insular regions (Figure 4d,e and Figure 5). Indeed, all *Kalanchoe* taxa occurring in the wild except for *K. gastonis-bonnieri* and *K. mortagei* had naturalized occurrences in the Andean Region; three of them (*K. densiflora*, *K. laxiflora,* and *K*. ×*houghtonii*) had also invasive occurrences (Figure 4d and Figure 5). A very high number of records of *K. laxiflora* (*N* = 123) and *K. densiflora* (*N* = 37) were reported as naturalized or invasive in the north part of the Andean Region (Pichincha Province) (Figure 5 b–d). In addition, almost all naturalized (20 out of 22) and all invasive records of *K*. ×*houghtonii* were located in the Andean Region. We detected four taxa (*K. gastonis-bonnieri*, *K. pinnata*, *K. tubiflora,* and *K.* ×*houghtonii*) with records categorized as naturalized in the Insular Region; of these, two species (*K. pinnata* and *K. tubiflora*) had also records categorized as invasive (Figure 4e and Figure 5). Notably, nearly all invasive occurrences of *K. pinnata* and *K. tubiflora* of Ecuador were detected in the Galapagos Islands (Figure 5).

We detected the presence of up to eleven *Kalanchoe* taxa inside PAs in Ecuador (Table 2). The Galapagos National Park was the PA with more taxa detected (*N* = 4). Furthermore, the taxon with more occurrences inside the PAs were *K. pinnata* (*N* = 32) in the Galapagos National Park and *K. laxiflora* (*N* = 18) in the Pululahua Geobotanical Reserve (Table 2). The four taxa detected inside the Galapagos National Park (*K. gastonis-bonnieri*, *K. pinnata*, *K. tubiflora*, and *K.* ×*houghtonii*) had at least one wild record (either as naturalized or invasive; Table 2). In addition, three species were found in the Pululahua Geobotanical Reserve (Andean Region), where *K. densiflora* was naturalized, *K. laxiflora* had several records as invasive, and *K. pinnata* had one record as uncertain in the wild (Table 2). *Kalanchoe laxiflora* was also reported as casual in Bellavista, a Private Protected Area located in the Andean Region. In the Cayambe Coca National Park (Andean Region), *K. blossfeldiana* and *K.* ×*houghtonii* were detected as cultivated and naturalized, respectively (Table 2). Finally, we were able to detect records of *K. pinnata* in two PAs in the Amazonian Region: as cultivated and escaped (but with an uncertain status) in the Yasuní National Park, and as cultivated but also fully naturalized in the Cuyabeno Lagartococha Yasuní Wetland Complex (Table 2).

### 2.5. Environmental Preferences

The PCA showed that the wild *Kalanchoe* records in the Galapagos Islands differ from those in the continent in their environmental conditions (Figure 6). These differences were particularly evidenced when comparing these conditions for the taxa present in both the Galapagos and continental Ecuador ranges (i.e., *K. pinnata*, *K. tubiflora*, and *K*. ×*houghtonii*) (Appendix A). The occurrences in the Galapagos Islands were associated with higher values of temperature and precipitation seasonality (bio4 and bio15, respectively), but lower precipitation in the wettest and coldest quarters (bio16 and bio19, respectively) (Appendix A). While in the Galapagos Islands the differences among taxa for the assayed environmental variables were rather low, we were able to find some large differences in the continent, and these mostly concerned *K. pinnata* and *K*. ×*houghtonii* (Appendix A). For the continent, in general terms, *K. pinnata* occurrences were more associated with higher temperatures (bio6) and precipitation (bio16 and bio19) than the other taxa, but also a higher temperature seasonality (bio4), a lower temperature annual range (bio7), and a lower precipitation seasonality (bio15) (Appendix A). The hybrid *K*. ×*houghtonii*, in contrast, showed lower values of bio4, bio16, and bio19 compared to many other *Kalanchoe* taxa. On a national scale, *K. pinnata* occurred in a wide range of temperature (bio4, bio6, and bio7) and precipitation (bio15, bio16, and bio19) conditions (Appendix A).

Regarding the human footprint (HF), comparisons between the Galapagos Islands and the continent (and among the taxa within the islands) were not possible because there are no assigned values of HF for the Galapagos Islands in the original raster file. For continental Ecuador, the differences were significant for several pairwise comparisons, with *K. densiflora*, *K. tubiflora*, and *K.* ×*houghtonii* being the taxa occurring in the areas with the highest anthropogenic impacts, and *K. pinnata* the one showing the widest range of HF values (Appendix A). The analysis of the anthropogenic biomes (AB) showed that *Kalanchoe* occurrences were more frequently located in villages, followed by dense settlements and rangelands (Appendix A). *Kalanchoe* plants, in contrast, rarely occurred in natural and seminatural habitats (wildlands and seminatural lands; Appendix A) and the few available records for these two biomes were almost exclusively in the Galapagos Islands (Appendix A). It is also relevant to mention that *K. pinnata* was the only species present in all anthromes (Appendix A).

## 3. Discussion

### 3.1. Kalanchoe Taxa in Ecuador

This study detected 16 *Kalanchoe* taxa in Ecuador, either wild or cultivated; compared with the previous knowledge [42,43,44,45,47], we are reporting two new *Kalanchoe* taxa for the Galapagos Islands (*K. carnea* and the hybrid *K.* ×*houghtonii*) and nine new taxa for continental Ecuador: *K. densiflora*, *K. fedtschenkoi*, *K. gastonis-bonnieri*, *K. laxiflora*, *K. mortagei*, *K. thyrsiflora*, *K. tomentosa*, *K. tubiflora*, and *K.* ×*laetivirens*. Among the new taxa reported, several have wild occurrences already naturalized (*K. densiflora* and *K. laxiflora*), whereas a previously cited species under cultivation status is now confirmed as occurring in the wild, also with well-established populations in the Galapagos Islands (*K. gastonis-bonnieri*). These three species, together with the five species previously reported in the literature in the wild in Ecuador (which are all naturalized except *K. mortagei*), should, thus, be regarded as constituents of the flora of Ecuador and, for this reason, we provide an identification key for them (see above). *Kalanchoe mortagei*, although at present only a casual alien species in Ecuador, in other places of tropical America it is already naturalized, e.g., in Cuba [35]. In contrast, *Kalanchoe crenata*, despite that in *Flora of Ecuador* [47] it was considered as one of the three species naturalized in the country, it has likely not escaped into the wild yet; the collection in which the naturalization status is supported (Dodson and Dodson 11727) seems to be from cultivated individuals (http://legacy.tropicos.org/Specimen/3009462; accessed on 22 April 2022). In addition, the other records we have gathered for this species are also of cultivated plants (Appendix A). We need to understand why *K. densiflora* and *K. laxiflora*, which are widely distributed along the Andean Region of Ecuador (both can be found between 500 and over 3000 m above sea level, a.s.l.) with some very large populations, had remained unnoticed until now. A reason is that our survey has not only included the “standard” information sources (i.e., herbaria and scientific publications) but also citizen science portals (such as iNaturalist), blogs, and other non-scientific websites (e.g., photo-sharing portals), which have proven to be very valuable sources of plant occurrences in recent times. These “non-standard” sources, for example, provided nearly 40% of the total occurrences of *K*. ×*houghtonii* in a recent study for defining its geographical distribution range at a global scale [39]. Another example is the first observation of *Noccaea perfoliata* (= *Microthlaspi perfoliatum*) for Texas thanks to an iNaturalist record [48].

Alien taxa of *Kalanchoe* are perceived as valuable for ornamental trade [49,50]. Indeed, *Kalanchoe* is ranked as the second most popular potted plant genus in Europe [50], most likely because its various taxa are beautiful plants (with often eye-catching flowers), easy to grow (they are drought-tolerant plants and do not demand much attention), and easy to propagate (many taxa have the ability to produce plantlets along leaf margins) [20]. The trade of ornamental plants is one of the main introduction pathways of invasive alien plants, if not the most important [10,11,51,52]. A recent study by Arianoutsou et al. [53] revealed that the main pathways of alien plants’ introduction into Europe are linked to accidental escapes from ornamental and horticultural activities. Furthermore, on a global scale, the escape of plants associated with horticultural purposes is the most important pathway subcategory of alien plant introductions [54]. In Ecuador, all *Kalanchoe* taxa that have been observed in the wild are also cultivated ornamental plants in private/public gardens or nurseries, suggesting that the presence of *Kalanchoe* taxa in the wild is mainly the result of escapes from nearby ornamental gardens.

Besides the fact that the main pathway for the introduction of *Kalanchoe* taxa is ornamental trade, the propagation success of *Kalanchoe* is also likely associated with its popularity as medicinal plants [20,55]. In Latin America, *Kalanchoe* taxa are used in folk medicine as anti-inflammatory, wound healing, insecticide, antibacterial, antioxidant, as well as for their alleged anti-cancer properties [20,56,57]. Of the taxa reported in Ecuador, at least two of them can be easily acquired from nurseries, garden centers, or online (for example, in popular e-commerce websites such as *Mercado Libre*) for their medicinal use. For example, *K. pinnata*, locally known as *planta milagrosa* (“miracle plant” in English), is used as anti-inflammatory in the Coastal Region of Ecuador; in addition, there are up to 18 therapeutic uses registered for this species in the Amazonian Region [58,59,60]. The leaves of *K. gastonis-bonnieri* (known in Ecuador as *dulcamara*, a name similar to that of *Solanum dulcamara*, and probably in an attempt to get more popularity as a medicinal plant) are used as a treatment for anemia, kidney stones, and as a topical analgesic in the Coastal Region [61], while in the Andean Region the leaves are used to cure cancer [62]. Furthermore, a recent study cites a total of 12 therapeutic uses throughout continental Ecuador, including the treatment of sexually transmitted diseases [63]. In Ecuador, besides the traditional medicinal uses, there are studies pointing out *K. gastonis-bonnieri* as a nutraceutical beverage [64,65] and feed for fattening broiler chickens [66,67].

We have been able to detect six *Kalanchoe* species (*K. blossfeldiana*, *K. carnea*, *K. crenata*, *K. eriophylla*, *K. fedtschenkoi*, and *K. tomentosa*) that occur in Ecuador exclusively as cultivated plants; that is, we did not find evidence of escape from gardens. At a global scale, with the exceptions of *K. crenata* and *K. fedtschenkoi*, these species do very rarely escape from cultivation (Descoings, 2003). This would also be the case for *K. thyrsiflora*, one of the two taxa having all their occurrences classified as uncertain; the two records we gathered are iNaturalist observations, but the pictures provided by the observers do not enable us to make sure that the plants are cultivated. While these two records would be cultivated, the two uncertain occurrences of *K.* ×*laetivirens*—which are also iNaturalist observations—would be, instead, wild ones, as this taxon has been observed out of cultivation in other tropical or subtropical regions of the world, such as Cuba [35], Hainan Island in China (J. López-Pujol, pers. obs.), and the Canary Islands [68]. Of the eight taxa with wild records in Ecuador, *K. mortagei* seems to be the less problematic at present, as it occurs as casual in the Amazonian Region (at about 500 m a.s.l.) and as uncertain in the wild in the Andes (at ca. 2700 m a.s.l.). This species has been reported as an alien species in the wild in Atlantic Forest in Sergipe (north-eastern Brazil; [69]) and in Honduras [70], and as naturalized in the locality of Cabo Cruz in Cuba [35]. We report a record of *K. gastonis-bonnieri* as naturalized in the Galapagos Islands (Isabela Island, near sea level); however, we could not find reliable information about the invasive potential of this species in South America in the literature, so we are reporting for the first time this species for this continent; it is, however, showing signs of escaping in other parts of the world, including Australia (https://www.inaturalist.org/observations/43488774; accessed on 22 April 2022), Hong Kong of China (https://www.inaturalist.org/observations/28761346; accessed on 22 April 2022), Puerto Rico (https://www.inaturalist.org/observations/103269505; accessed on 22 April 2022), and Spain [71].

*Kalanchoe pinnata*, in contrast, is a real threat as it is naturalized in the four regions of Ecuador, it is already present in four PAs, and is invasive in the Galapagos Islands where it was introduced 120 years ago (the oldest record of *K. pinnata* is of 1905, from San Cristóbal Island). Originally endemic to Madagascar, it seems that it was already present in mid-1800s in western Africa, when it was brought to Europe [72]. The species would have also reached, more sooner than later, the Americas, as it was present in Cuba around 1875 [35]. Our study shows that *K. pinnata* occurs in a wide range of temperatures and precipitations in Ecuador (as well as in a wide range of elevations, between sea level to approximately 3000 m a.s.l.), suggesting the high capacity of this species to acclimate to a wide variety of conditions. This is likely one of the main reasons to explain why today *K. pinnata* is the most widespread species of the genus, as it has become naturalized in mild-climate areas throughout the world [72] and even been reported as invasive in several countries [19,73,74,75,76,77,78]. In the Galapagos Islands, *K. pinnata* has been considered one of the most aggressive invasive plant species, having already invaded large areas [43,79,80,81]. It is the only *Kalanchoe* species naturalized in the Ecuadorian Amazon. Since the 1970, *K. pinnata* has been reported as the most common species along much of the Old Trail in the transition zone on Santa Cruz Island in the Galapagos Islands [82]. The effective control of this species is complicated in the Galapagos Islands due to its active regeneration after eradication by propagule bank in the soil [81]. A study in a seasonally dry tropical forest in Veracruz (Mexico) reported that *K. pinnata* invasion is characterized by both sexual and vegetative reproduction. This strategy allows the species to form dense stands that diminishes richness, individual abundance, and recruitment of indigenous plant species, and increases the organic matter and total carbon pools in the soil [83]. An experimental study demonstrated allelopathic effects caused by aqueous extracts of foliage/roots of *K. pinnata* on germination and seedling growth of some crop plant species [84]. Based on the invasion records of *K. pinnata* in Ecuador and the background literature, we propose that *K. pinnata* should be considered an invasive species on a national scale, thus requiring surveillance and containment of its populations.

Two of the newly reported species for the country, *K. densiflora* and *K. laxiflora*, are behaving as invasive in the Andean Region. The oldest records of *K. densiflora* and *K. laxiflora* in Ecuador are in this region; this, combined with the fact that very few wild occurrences are out of the Andes, suggest that the first introduction of both species took place in the *cordillera*. *Kalanchoe densiflora* and *K. laxiflora* have similar environmental preferences (Figure 6 and Appendix A) and, despite being widely distributed in the Ecuadorian Andes, most naturalized and invasive occurrences are concentrated in the Pichincha Province (Figure 5b,d). Both species would have escaped from gardens in the urban areas of this province that is home of the capital district (Quito), the most populated metropolitan area of Ecuador with nearly 3 million people. *Kalanchoe laxiflora* has already invasive records in the PA of Pululahua Geobotanical Reserve, located 25 km north of Quito. However, little is known about the invasive potential of *K. densiflora* and *K. laxiflora* in other parts of the world. *Kalanchoe densiflora* has been reported as invasive in several areas near Bogotá (Colombia), in xeric scrub, especially in rocky soil areas [85] and in urban and peri-urban areas [86]. Regarding *K. laxiflora*, we could not find previous reports of this species as invasive, but we are aware of naturalized or casual occurrences, including Bolivia (https://www.gbif.org/es/occurrence/1260609218; accessed on 22 April 2022; https://www.gbif.org/es/occurrence/2595700729; accessed on 22 April 2022) and Spain (that was mistakenly identified as *K. fedtschenkoi* [87]). Future studies evaluating the impacts of these species on the structure and function of the Andean ecosystems are required to assess whether these species should be considered a management priority.

The hybrid *Kalanchoe* ×*houghtonii* and its parental species (*K. daigremontiana and K. tubiflora*) are extensively naturalized, and sometimes invasive (particularly the hybrid and *K. tubiflora*) in many countries [19,39,88]. In Ecuador, the less problematic of the three taxa is *K. daigremontiana*, which is naturalized at a single location in the Andean Region (in northern Pichincha Province, at an elevation of about 2500 m a.s.l.). *Kalanchoe* ×*houghtonii* and *K. tubiflora* are naturalized in several locations along the Andes (where they can reach elevations of nearly 4500 and 3000 m a.s.l., respectively) and the Galapagos Islands (near sea level), with *K*. ×*houghtonii* being invasive in the first (in Pichincha Province) and *K. tubiflora* being invasive in the second (it is already invading the Galapagos National Park). The three taxa have a very efficient clonal reproduction; asexually produced plantlets grow in the margins of leaves that establish by dropping to the ground, tending to form a “carpet” of plantlets on the soil [23,89]. In Queensland (Australia), *K. tubiflora* has become a severe threat to the grazing industry [19]; the species is toxic to livestock and inhibits the pasture growth throughout allelopathic root exudates [31,90,91]. In an arid zone in Venezuela, *K*. ×*houghtonii* (identified erroneously as *K. daigremontiana*, see [39]) is invading Cerro Saroche National Park, where the taxon reduces indigenous cacti recruitment [92] and modifies the pool and mineralization of nitrogen and carbon in the soil [93]. There are no studies in Ecuador that estimate the impact caused by *K. tubiflora* and *K.* ×*houghtonii* yet.

The environmental conditions where wild *Kalanchoe* taxa are dwelling are considerably different between continental Ecuador and the Galapagos Islands, indicating the high capacity of these taxa to acclimate to a wide variety of climatic conditions, as well as across very different degrees of human disturbance (i.e., from almost pristine habitats, particularly in the Galapagos Islands, to heavily urbanized ones). This high acclimation ability is probably linked to the fact that most taxa of the genus are drought-tolerant plants, can tolerate near-freezing temperatures but also high ones (about 40 °C), are able to grow from full sun to partial shade and, also importantly, have no soil requirements, being able to grow for example on bare rocks or on the concrete [20,72]; as a pertinent example, *K*. ×*houghtonii* was one of the first plants colonizing the new habitats formed during the volcanic eruption in La Palma (Canary Islands, Spain) in late 2021 (https://www.ibb.csic.es/es/2022/01/kalanchoe-houghtonii-hibrido-invasor-amenaza-colonizar-colada-volcanica-la-palma/; accessed on 20 June 2022). In addition to *Kalanchoe* ecophysiology, the genus has a very efficient propagation mode that combines the sexual reproduction with a range of different asexual reproductive strategies (such as the well-known formation of plantlets on leaf margins, but also the formation of bulbils on inflorescence nodes [94]) that may help the plant to survive on changing habitats and niches. The environmental differences detected between the *Kalanchoe* occurrences of continental Ecuador and those of the Galapagos Islands could indicate differentiation in the realized niche, challenging future invasion predictions based on climatic variables. As the popular species distribution models are based on the principle of niche conservatism, assessing possible niche shifts for *Kalanchoe* taxa between the continent and the islands is a prerequisite to improving our capacity to prevent future invasions.

### 3.2. Suggestions for Management

Potential biocontrol agents, options of mechanic control and specific herbicides have been identified to control populations of invasive taxa of *Kalanchoe* (e.g., [19,24,95,96]). In Ecuador, however, invasive taxa of *Kalanchoe* have been considered as a management priority (and, thus controlled) only in the Galapagos Islands (e.g., [81]). Eradication is not always enough to ensure the effective management of a biological invasion; prevention strategies should play a central role, with identification of the main introduction pathways as a mandatory step to prevent an invasion [10]. Our results show that ornamental trade is the main pathway for the introduction of *Kalanchoe* taxa in Ecuador, highlighting the importance of reducing the ornamental value and use of alien taxa with invasive potential in horticulture and, at the same time, promoting the use of indigenous species. Additionally, the present study has allowed us to identify *Kalanchoe* taxa with a proven potential to invade specific regions of Ecuador. Specifically, the cultivation in gardens of (i) *K. daigremontiana*, *K. pinnata*, *K. tubiflora*, and *K.* ×*houghtonii* in the Galapagos Islands, (ii) *K. densiflora* and *K. laxiflora* in the Andean Region, and (iii) *K. pinnata* in the Ecuadorian Amazon could generate new biological invasions; therefore, the cultivation of these species should be avoided. Some *Kalanchoe* taxa with invasive potential are cultivated even within PAs, e.g., *K. gastonis-bonnieri* at the Galapagos National Park and *K. pinnata* at Yasuní National Park. This information can be used to generate a blacklist of *Kalanchoe* taxa by region and protected area. Blacklists are a strategy made up for alien species with very high risk of invasion that may produce negative impacts on indigenous biodiversity [97,98,99]. A blacklist, at the same time, includes three sub-lists, according to the available distribution and eradication measures [100]. First, a Blacklist–Warning List includes invasive alien species not yet present in the reference area. Second, a Blacklist–Action List consists of invasive alien species present in a few localities, and control/eradication measures are possible, reducing the probability of spread. Lastly, a Blacklist–Management List comprises invasive alien species in a few localities or widely distributed with hardly any feasible eradication measures. Finally, science outreach and education are required to change people’s perspective on *Kalanchoe* from valuable plants to noxious taxa. 

## 4. Materials and Methods

### 4.1. Study Area

The study area includes the entire territory of the Republic of Ecuador located in the northwest of South America, between Colombia, Peru, and the Pacific Ocean (Figure 7). From west to east, Ecuador has four biogeographic regions, well defined by the Andes Mountains and the Pacific Ocean: The Insular Region (Galapagos Islands), the Coastal Region, the Andean Region, and the Amazonian Region. The Galapagos Islands are a volcanic archipelago of less than 8000 km^2^ located ca. 900 km off the Pacific coast of Ecuador. Continental Ecuador (ca. 250,000 km^2^) has a very complex topography, with peaks exceeding 5000 m a.s.l. The impressive diversity of ecosystems (from the desert-like coastal lowlands to the very cold Andes summits and the Amazonian rainforests) is one of the main reasons that explains why Ecuador is a megadiverse country; it harbors about 17,500 vascular plant species, of which 5500 are endemic to the country [101].

### 4.2. List of Kalanchoe Taxa and Occurrences

For the inventory of *Kalanchoe* taxa in Ecuador, we compiled the records of any taxa belonging to this genus (incl. *Bryophyllum*; see Supplementary Text S1 for taxonomical details) from an extensive search—carried out during the period of 2020–2022—using various information sources: (1) specimens from the National Herbarium of Ecuador (QCNE) and other national and international herbaria, both offline and online (e.g., through Tropicos [https://www.tropicos.org], accessed on 1 March 2020–31 October 2021); (2) global and national biodiversity databases, including the Global Register of Introduced and Invasive Species (GRIIS) [42], the Catalogue of Vascular Plants of Ecuador (W3CEC) (http://legacy.tropicos.org/Project/CE; accessed on 1 March 2020–31 October 2021), the Global Biodiversity Information Facility (GBIF; https://www.gbif.org/country/EC/summary; accessed on 1 March 2020–31 October 2021), and the iNaturalist citizen science web portal (https://www.inaturalist.org/; accessed on 1 March 2020–31 October 2021); (3) scientific publications (articles and monographs) searched through scholarly web browsers and databases (ISI web of knowledge, SCOPUS, ScieELO, and Google Scholar) using the descriptors “*Kalanchoe*” or “*Bryophyllum*” and “Ecuador”; (4), PhD/Master dissertations and project reports from repositories of public and private institutions in Ecuador; (5) personal blogs and other non-scientific websites (e.g., Flickr); and (6) personal observations. 

For each record we registered: (1) taxon name, (2) collector or observer, (3) geographic coordinates, (4) locality, (5) year of collection or observation, (6) whether they have associated images. All gathered records were thoroughly validated one by one. Given the high rate of identification errors in the genus (up to nearly 50% in some cases; [39]), we only kept those we directly observed (as field observations or as checked herbarium specimens) or those for which an image was available (e.g., pictures showing living individuals or scanned herbaria sheets). Using botanical descriptions of the *Kalanchoe* taxa [19,20,21,35,40,72,102,103,104,105] and our own knowledge, we analyzed each record to verify its taxonomic identification and we corrected any erroneous identification when necessary. We excluded the records with confusing identifications, no date or no georeferencing information. Additionally, we elaborated a botanical key for the taxonomic identification of *Kalanchoe* taxa occurring in the wild (i.e., outside cultivation) in Ecuador.

### 4.3. Spatial–Temporal Distribution of the Records of Kalanchoe Taxa

We took the oldest year of collection or observation as a conservative estimate of the year of the first introduction for each taxon occurring in Ecuador. To examine whether the number of observations has increased over time, we constructed a record–accumulation curve over time for each taxon, using R v. 4.1.1 [106]. To examine the spatial distribution of the occurrences, we created record distribution maps in cumulative periods of 20 years. All maps were produced using QGIS 3.20 Odense. 

To check whether there are occurrences of *Kalanchoe* taxa within the Ecuadorian protected areas (PAs), the digitized map of PAs [107] was overlapped with the taxa occurrences. For the Galapagos Islands, specifically, a land-use layer was overlapped to discard the occurrences of *Kalanchoe* taxa in urban areas [108]. The intersect tool of QGIS was used to estimate the number of occurrences of *Kalanchoe* taxa located within PAs.

### 4.4. Categorization of Kalanchoe Taxa by Invasion Status

To categorize the *Kalanchoe* taxa according to their invasion status, we first classified all valid *Kalanchoe* records into three categories (i) wild occurrences (=occurrences outside cultivation), (2) cultivated occurrences, and (3) uncertain ones (i.e., if we were not sure that a record was cultivated or outside cultivation). Then, the records categorized as wild went through the next level of the classification: (1) casual, when our observations, the images, or the information contained in the description of each record suggested that the individuals of that location do not form self-replacing populations (see definitions regarding invasion status in [109]); (2) naturalized, when they apparently form self-replacing populations without direct human intervention for a period of at least 10 years; in practice, records were regarded as naturalized when pictures showed individuals in different developmental stages (seedlings, young adults, and mature individuals), (3) invasive, when self-replacing populations are very large and are apparently damaging the recipient ecosystem; and (4) uncertain (in the wild). Finally, all identified *Kalanchoe* taxa occurring in the wild in Ecuador were classified as casual, naturalized, and invasive when all their occurrences were casual, and at least some were naturalized and invasive, respectively.

### 4.5. Environmental Preferences

In order to describe the environmental conditions in which each taxon of *Kalanchoe* occur in the wild in Ecuador, we gathered the values of several variables of all records; these included the standard 19 bioclimatic variables, the human footprint (HF), and the anthropogenic biomes (AB). The latter two variables were chosen because the establishment and spread of *Kalanchoe* taxa is very often related to human disturbance. The bioclimatic variables were obtained from the WorldClim database (http://www.worldclim.org/; accessed on 14 February 2022). To avoid model overfitting, we performed a Pearson correlation analysis to choose only those variables with low correlation with each other (<|0.7|). Selection of variables from groups of highly correlated ones was based on expert criteria of the taxa ecology and the biological significance of curve responses of presence and background points (Appendix A, in supplementary material). The bioclimatic variables finally selected were: (i) temperature seasonality (bio4), (ii) minimum temperature of coldest month (bio6), (iii) temperature annual range (bio7), (iv) precipitation seasonality (bio15), (v) precipitation of wettest quarter (bio16), and (vi) precipitation of coldest quarter (bio19). The HF was taken from Sanderson [110]; it is a variable based on the anthropogenic impacts on the environment and was created from nine global data layers covering various aspects of human influence (population density, human land use, and infrastructure and human access). While HF and the bioclimatic variables are continuous, the AB of the world are a categorical variable, which was obtained from Ellis et al. ([111], version 2). In short, the AB scheme is the characterization of terrestrial biomes into new categories based on their direct human disturbance. To ease the analyses, we used six simplified anthrome levels: (i) dense settlements, (ii) villages, (iii) croplands, (iv) rangelands, (v) seminatural, and (vi) wildlands. Values of each variable (and the type of AB) for each *Kalanchoe* record in the wild were extracted, and Pearson correlation was carried out, in R v. 4.1.1 [106] using the RStudio platform [112]. 

Descriptive statistics for each variable were computed to define the environmental conditions where each taxon inhabits in each of the two main ranges in Ecuador: Galapagos Islands and continental Ecuador. Comparisons among groups (i.e., among taxa but also between the two geographic ranges for each of the taxa) were statistically tested. First, the Shapiro–Wilk test was performed to check the normality of the data; as they were not normally distributed (*p*-value was less than alpha level, 0.05), the non-parametric analysis Wilcoxon–Mann–Whitney was conducted. The analyses were performed using the shapiro.test and wilcox.test functions of stats package v. 4.1.1 in R [106]. Finally, we used a principal component analysis (PCA) to examine the environmental variability of the realized niches (i.e., using all occurrences) of *Kalanchoe* taxa in the wild in Ecuador, by distinguishing the records of the Galapagos Islands from these of the continent. The PCA was run with the HF and the six bioclimatic variables also in R.

## Figures and Tables

**Figure 1 plants-11-01746-f001:**
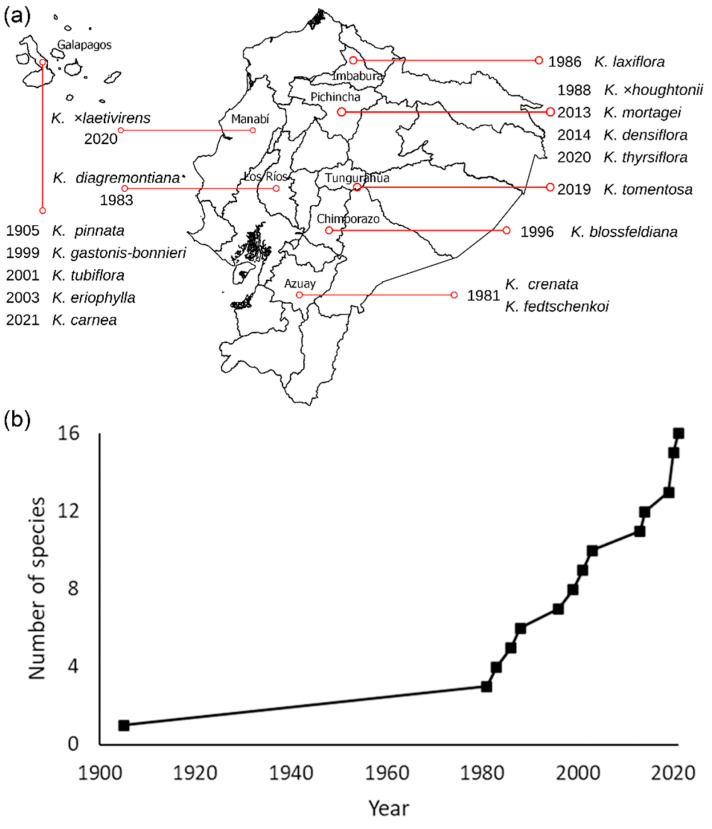
(**a**) First year and location of collection or observation for each *Kalanchoe* taxon reported in Ecuador (the names of Ecuadorian provinces where taxa of *Kalanchoe* were firstly observed are indicated); (**b**) Taxa accumulation curve through time.

**Figure 2 plants-11-01746-f002:**
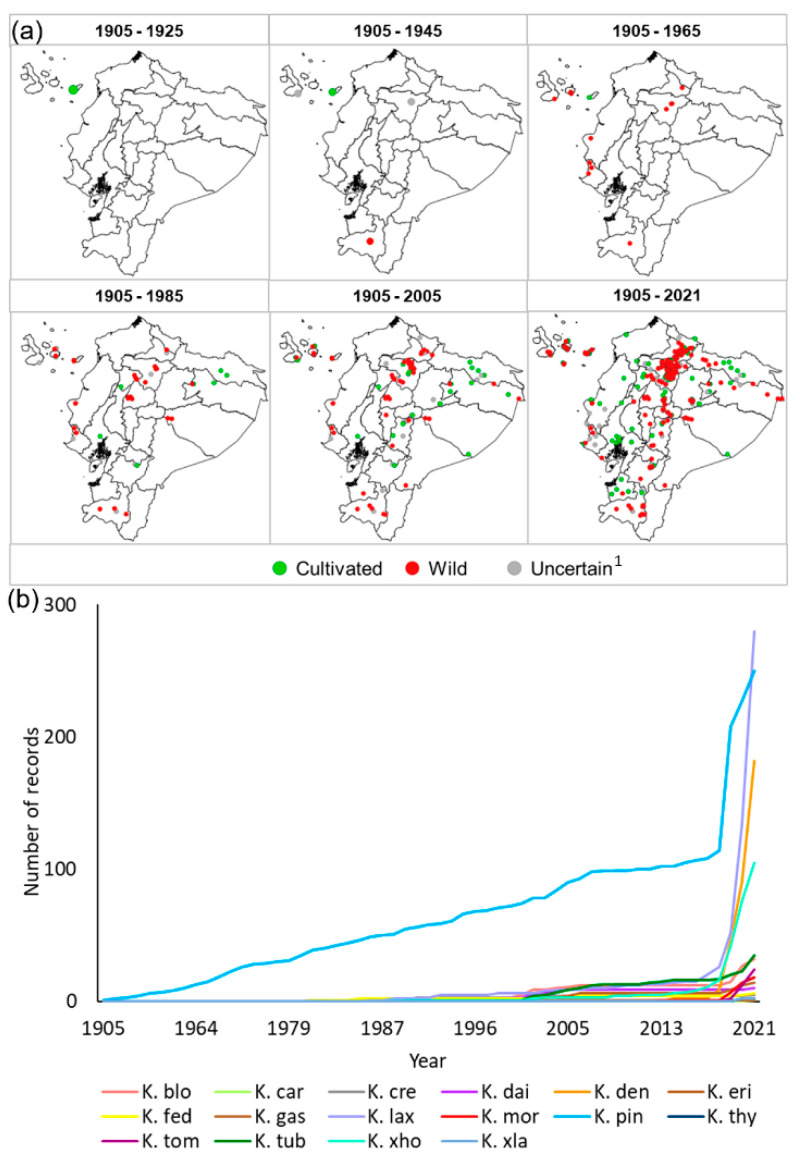
Spatial and temporal accumulation of *Kalanchoe* records in Ecuador, categorized according to the possible occurrence status (cultivated, wild, and uncertain) (**a**) and temporal record accumulation for each *Kalanchoe* taxon (**b**). Taxa codes are as follows: K. blo: *K. blossfeldiana* (*N* = 32), K. car: *K. carnea* (*N* = 1), K. cre: *K. crenata* (*N* = 5), K. dai: *K. daigremontiana* (*N* = 10), K. den: *K. densiflora* (*N* = 182), K. eri: *K. eriophylla* (*N* = 1), K. fed: *K. fedtschenkoi* (*N* = 6), K. gas: *K. gastonis-bonnieri* (*N* = 14), K. lax: *K. laxiflora* (*N* = 280), K. mor: *K. mortagei* (*N* = 18), K. pin: *K. pinnata* (*N* = 250), K. thy: *K. thyrsiflora* (*N* = 2), K. tom: *K. tomentosa* (*N* = 24), K. tub: *K. tubiflora* (*N* = 35), K. ×ho: *K*. ×*houghtonii* (*N* = 105), and K. ×la: *K*. ×*laetivirens* (*N* = 2). ^1^ “Uncertain” means that the records cannot be categorized either as wild or cultivated.

**Figure 3 plants-11-01746-f003:**
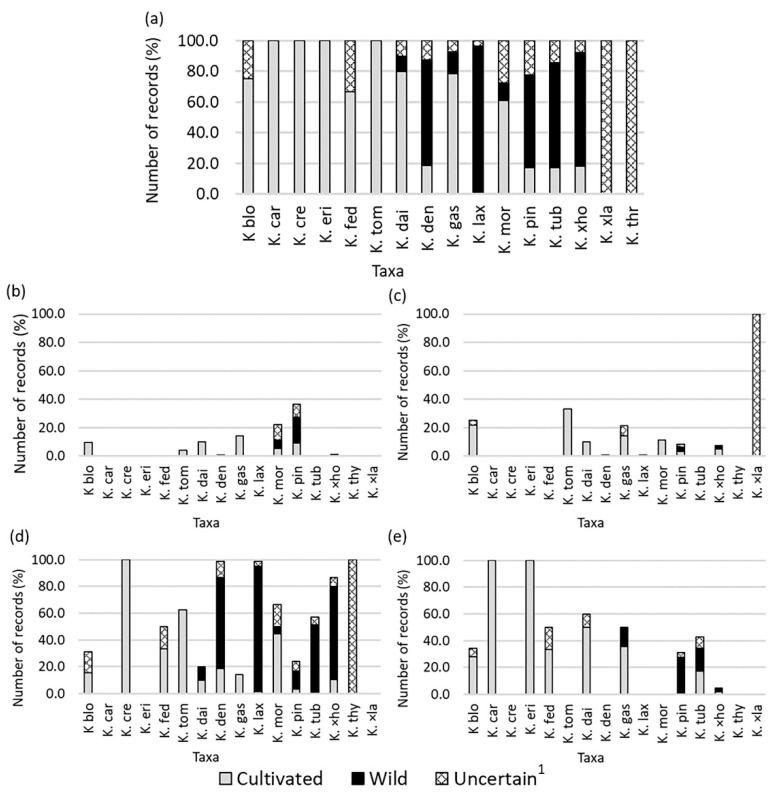
Number of records (%) per *Kalanchoe* taxa categorized as cultivated, wild, or uncertain for the whole Ecuador (**a**), for the Amazonian Region (**b**), for the Coastal Region (**c**), for the Andean Region (**d**), and for the Insular Region (**e**). Taxa codes are as follows (with record numbers shown in parenthesis for cultivated, wild, and uncertain ones, in this order): K. blo: *K. blossfeldiana* (*N* = 32 (32/0/0)), K. car: *K. carnea* (*N* = 1 (1/0/0)), K. cre: *K. crenata* (*N* = 5 (5/0/0)), K. dai: *K. daigremontiana* (*N* = 10 (8/1/1)), K. den: *K. densiflora* (*N* = 182 (34/125/23)), K. eri: *K. eriophylla* (*N* = 1 (1/0/0)), K. fed: *K. fedtschenkoi* (*N* = 6 (4/0/2)), K. gas: *K. gastonis-bonnieri* (*N* = 14 (11/2/1)), K. lax: *K. laxiflora* (*N* = 280 (4/264/12)), K. mor: *K. mortagei* (*N* = 18 (11/2/5)), K. pin: *K. pinnata* (*N* = 250 (43/152/55)), K. thy: *K. thyrsiflora* (*N* = 2 (0/0/2)), K. tom: *K. tomentosa* (*N* = 24 (24/0/0)), K. tub: *K. tubiflora* (*N* = 35 (5/6/24)), K. ×ho: *K*. ×*houghtonii* (*N* = 105 (19/78/8)), and K. ×la: *K*. ×*laetivirens* (*N* = 2 (0/0/2)). ^1^ “Uncertain” means that the records cannot be categorized either as wild or cultivated.

**Figure 4 plants-11-01746-f004:**
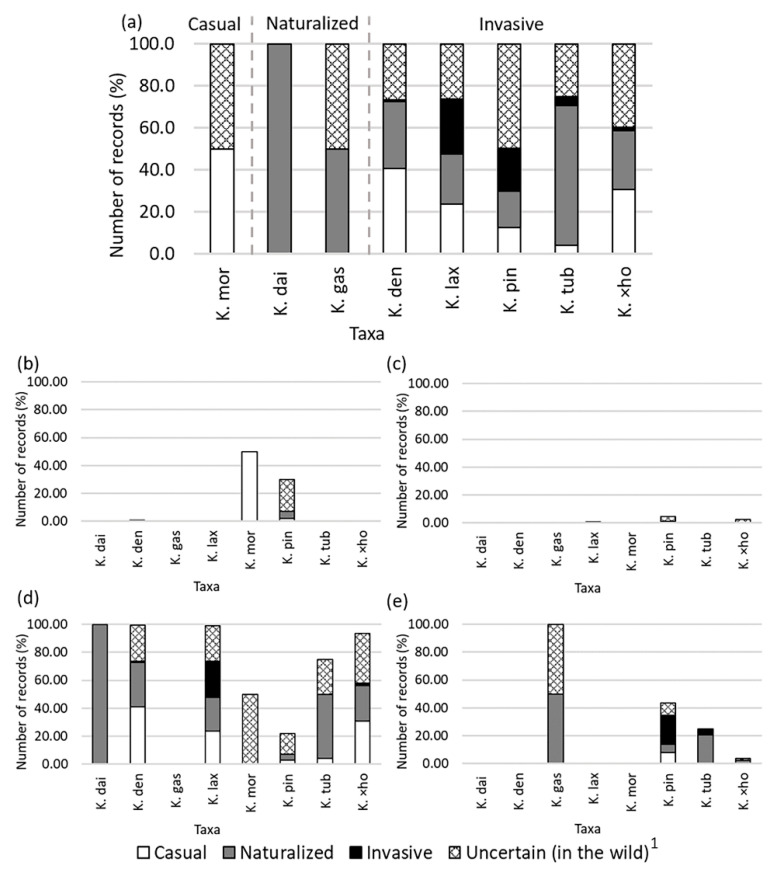
Number of records in the wild (%) per *Kalanchoe* taxa categorized as casual, naturalized, invasive, or uncertain for the whole Ecuador (**a**), for the Amazonian Region (**b**), for the Coastal Region (**c**), for the Andean Region (**d**), and for the Insular Region (**e**). Taxa codes are as follows (with record numbers shown in parenthesis for casual, naturalized, invasive, and uncertain ones, in this order): K. dai: *K. daigremontiana* (*N* = 1 (0/1/0/0)), K. den: *K. densiflora* (*N* = 125 (51/40/1/33)), K. gas: *K. gastonis-bonnieri* (*N* = 2 (0/1/0/1)), K. lax: *K. laxiflora* (*N* = 266 (63/64/69/70)), K. mor: *K. mortagei* (*N* = 2 (1/0/0/1)), K. pin: *K. pinnata* (*N* = 151 (19/26/31/75)), K. tub: *K. tubiflora* (*N* = 24 (1/16/1/6)), and K. ×ho: *K.* ×*houghtonii* (*N* = 78 (24/22/1/31)). ^1^ “Uncertain (in the wild)” are for records categorized as wild but we have no certainty on its invasion status (casual, naturalized, or invasive).

**Figure 5 plants-11-01746-f005:**
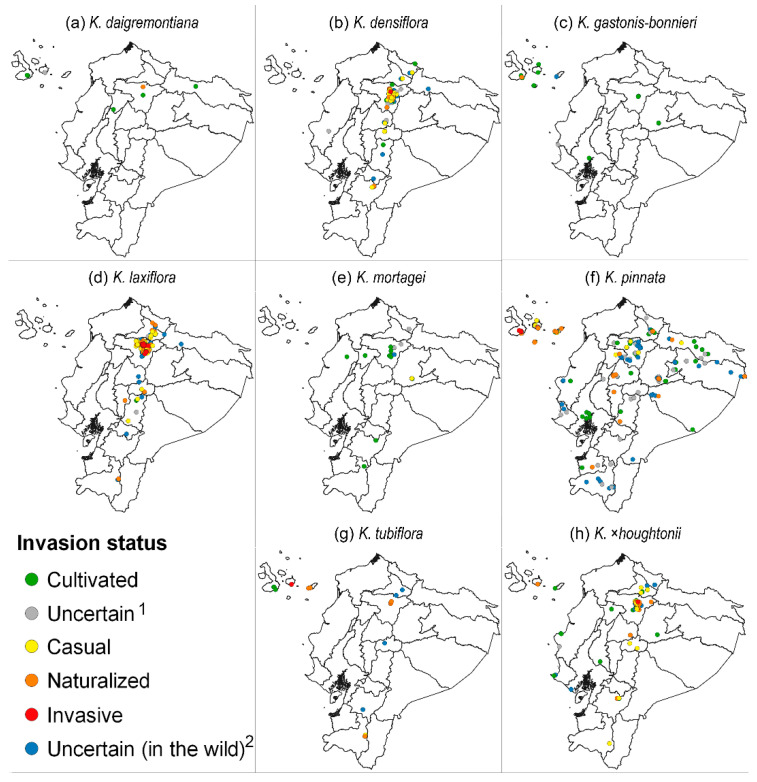
Geographic distribution of *Kalanchoe* taxa with occurrences in Ecuador, sorted by status: (**a**) *K. daigremontiana*, (**b**) *K. densiflora*, (**c**) *K. gastonis-bonnieri*, (**d**) *K. laxiflora*, (**e**) *K. mortagei*, (**f**) *K. pinnata*, (**g**) *K. tubiflora*, and (**h**) *K.* ×*houghtonii*. ^1^ “Uncertain” means that the records cannot be categorized either as wild or cultivated; ^2^ “Uncertain (in the wild)” are for records categorized as wild but we have no certainty on its invasion status (casual, naturalized, or invasive).

**Figure 6 plants-11-01746-f006:**
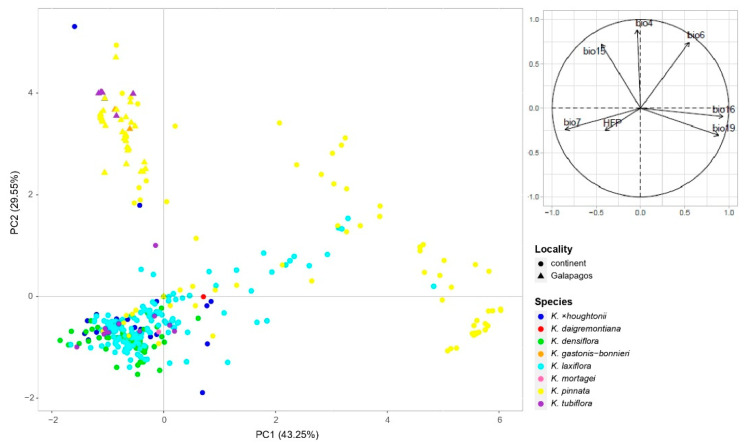
Principal Component Analysis (PCA) performed with bioclimatic variables (plus human footprint) of *Kalanchoe* taxa for their wild occurrences in the Galapagos Islands and continental Ecuador.

**Figure 7 plants-11-01746-f007:**
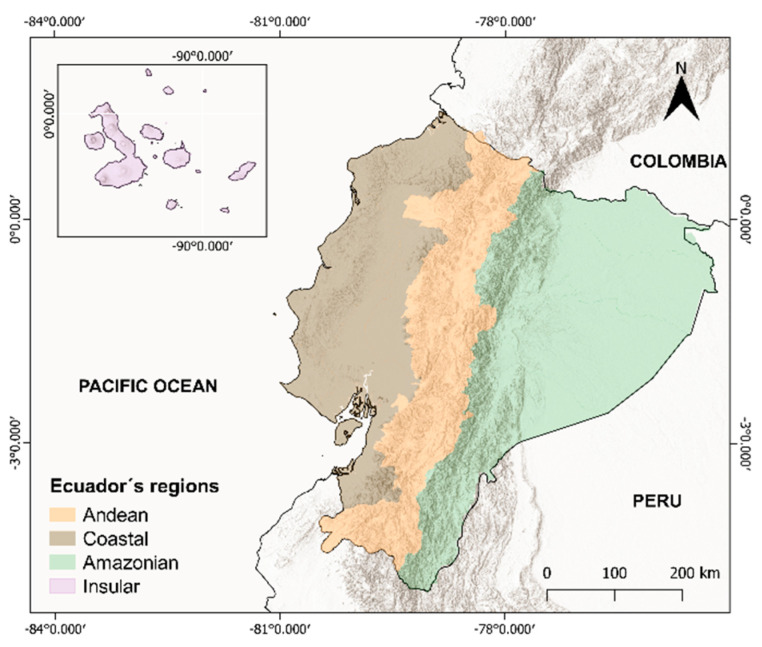
Map of the study area. The four biogeographic regions of Ecuador are shown.

**Table 1 plants-11-01746-t001:** Number of records of *Kalanchoe* taxa per bioregion (Amazonian, Andean, Coastal, and Insular regions) in Ecuador (including wild, cultivated, and uncertain^1^ records). In parentheses, number of (verified) wild and cultivated occurrences per taxa, respectively. ^1^ “Uncertain” means that the records cannot be categorized either as wild or cultivated.

*Kalanchoe* Taxa	Amazonian	Andean	Coastal	Insular	Total (%)
*K. blossfeldiana* Poelln.	3 (0/3)	10 (0/5)	8 (0/7)	11 (0/9)	32 (3.3%)
*K. carnea* N.E. Br.	0	0	0	1 (0/1)	1 (0.1%)
*K. crenata* (Andrews) Haw.	0	5 (0/5)	0	0	5 (0.5%)
*K. daigremontiana* Raym.-Hamet and H. Perrier	1 (0/1)	2 (1/1)	1 (0/1)	6 (0/5)	10 (1.0%)
*K. densiflora* Rolfe	1 (1/0)	180 (124/ 34)	1	0	182 (18.8%)
*K. eriophylla* Hils. and Bojer ex Tul.	0	0	0	1 (0/1)	1 (0.1%)
*K. fedtschenkoi* Raym.-Hamet and H. Perrier	0	3 (0/2)	0	3 (0/2)	6 (0.6%)
*K. gastonis-bonnieri* Raym.-Hamet and H. Perrier	2 (0/2)	2 (0/2)	3 (0/2)	7 (2/5)	14 (1.4%)
*K. laxiflora* Baker	1 (1/0)	277 (263/4)	2	0	280 (29.0%)
*K. mortagei* Raym.-Hamet and H.Perrier	4 (1/1)	12 (1/8)	2 (0/2)	0	18 (1.9%)
*K. pinnata* (Lam.) Pers.	91 (45/23)	60 (33/9)	21 (8/8)	78 (66/3)	250 (25.9%)
*K. thryrsiflora* Harv.	0	2	0	0	2 (0.2%)
*K. tomentosa* Baker	1 (0/1)	15 (0/15)	8 (0/8)	0	24 (2.5%)
*K. tubiflora* (Harv.) Raym.-Hamet	0	20 (18/0)	0	15 (6/6)	35 (3.6%)
*K*. ×*houghtonii* D.B. Ward	1 (0/1)	91 (73/11)	8 (2/5)	5 (3/2)	105 (10.9%)
*K*. ×*laetivirens* Desc.	0	0	2	0	2 (0.2%)
Total records (%)	105 (11%)	679 (70%)	56 (6%)	127 (13%)	967 (100%)

**Table 2 plants-11-01746-t002:** List of *Kalanchoe* taxa and record status inside protected areas. The number of records is shown in parenthesis. “Uncertain” means that the records cannot be categorized either as wild or cultivated, whereas “Uncertain (in the wild)” are for records categorized as wild but we have no certainty on its invasion status (Casual, Naturalized, or Invasive).

Protected Areas	Taxa	Records and Occurrence Status
Galapagos National Park	*K. gastonis-bonnieri*	Naturalized (1)
	*K. pinnata*	Casual (2); Naturalized (5); Invasive (21); Uncertain (in the wild) (4)
	*K. tubiflora*	Invasive (1)
	*K.* ×*houghtonii*	Naturalized (2)
Pululahua Geobotanical Reserve	*K. pinnata*	Uncertain (in the wild) (1); Uncertain (1)
	*K. densiflora*	Naturalized (4)
	*K. laxiflora*	Casual (1); Naturalized (1); Invasive (14); Uncertain (in the wild) (2)
Bellavista Private Protected Area	*K. laxiflora*	Casual (1)
Yasuní National Park	*K. pinnata*	Uncertain in the wild (2); Uncertain (1); Cultivated (1)
Cuyabeno–Lagartococha–Yasuní complex	*K. pinnata*	Naturalized (2); Uncertain (in the wild) (2); Cultivated (1)
Cayambe Coca National Park	*K.* ×*houghtonii*	Naturalized (1)
	*K. blossfeldiana*	Cultivated (1)

## Data Availability

Data used in the analysis are available in the Appendix A.

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
