# Peer review of "The Genus Kalanchoe (Crassulaceae) in Ecuador: From Gardens to the Wild"

_plants, 2022, doi:10.3390/plants11131746_

Round 1
Reviewer 1 Report
Reviewer comments on the manuscript entitled “The Genus Kalanchoe in Ecuador: from Gardens to Wildlife”
This is a very good and interesting paper and there is no reason why it should not be published.
The authors may, however, want to consider some of the comments I provide below. In this regard I would strongly advise the authors to reread the manuscript and to refine some of the statements contained in the document.
A few general comments:
I have not paid attention to the possibility that editorial refinements are required to have the manuscript conform to Plants policy. If there are discrepancies between the formatting of what was submitted and what you require of manuscripts, I am sure you and the authors will attend to such.
Also, I have not double checked the literature references (what is cited is in the list, and what is in the list is cited).
Other comments, in no particular order:
· The authors might want to briefly define, typically in the ‘Material & Methods’, what they mean by “wildlife”. For example, does “in the wild” refer to “[comparatively] undisturbed natural vegetation / habitats”—for the sake of the reader it will help to be exact about what is meant.
· Also, please check the heading of Table 1 “(verified) wild and cultivated taxa,”. The impression is created that (at least) some kalanchoes are indigenous to Ecuador. I am sure this is unintentional.
· I would suggest absolute accuracy throughout. For example, in the first line reference is made to “natural range”. I’d imagine that the authors refer to is “natural geographical distribution range”.
· Still in this vein, in the third line it is stated “…of concern”. It would be good to already here state to what “concern” is referred. I.e., “conservation concern”; “transformation of comparatively undisturbed natural vegetation / habitats”?
· In line 58 “thick foliage” can be understood in different ways. I suppose the authors in this instance mean “succulent leaves” rather than a dense, leafy plant canopy.
· This sentence (found in lines 61–62): “Many taxa of Kalanchoe reproduce both sexually and clonally, with the plantlets asexually produced growing in the margins of the leaves.” could perhaps be rewritten as “Many taxa of Kalanchoe reproduce both sexually and clonally, with the plantlets asexually produced being variously carried along the margins of the leaves or, typically post-anthesis, on the inflorescences.” I.e., inflorescence bulbils are also produced asexually. See also line 355.
· My preference is to refer to: “the genus Kalanchoe” rather than “the Kalanchoe genus”. (See line 64.)
· The sentence in line 72: “Kalanchoe plants are also well known for having caused cattle and household pets poisoning episodes [16,31–33].” I would recast slightly, as follows: “Kalanchoe plants are also well known for causing episodes of poisoning of cattle and domestic, mammalian pets [16,31–33].” Or is material of Kalanchoe also poisonous to birds kept as domestic pets?
· For what it’s worth: the term “native” is regarded as derogatory by some people. May I suggest that the word “indigenous” rather be used? For example when referring to the flora that occur naturally in a country / region / etc.
· Please carefully check the very last line of Table 1. As currently constituted it looks like the final “Total (%)” is “967”.
· In lines 314 and 420 “taxa” should be singular, I presume. And in those contexts, “species” rather than “taxon” might be preferable.
· Please consider throughout not having a space between the multiplication sign “×” and the initial letter of the epithet followed by it, in line with Rec. H3.A.1 of the Code.
· “Continent” or “subcontinent” in line 402?
· Is “[16]” the correct reference for South Africa in line 406? I somehow doubt it.
· In line 425 “reproductive” should be “sexual”.
· Always give a genus name in full when it is the first word of a sentence.
Reviewer 2 Report
Dear Authors,
I read the manuscript with great interest. I propose the following suggestions and corrections for manuscript improvement:
Title:
- I suggest to add the name of Crassulaceae (in parenthesis) after Kalanchoe
Key words:
- I suggest to change “alien plant species” and “biological invasions” into “biological records” and “invasive alien species”
Introduction
- page 3, line 103: I suggest to replace “degree of invasion” by “categories of alien plants”
Results
- Figure 5: Please, provide one word “Status” above all colored dots (delete Wildlife status) in the legend; change the explanation for a gray dot as “uncertain cultivation”, and for a blue dot as “uncertain alien status”
- it would be interesting to provide (if possible) more detailed information on vertical distribution of the studied taxa in Ecuador (maybe in subchapter of Environmental preferences or in Discussion)
Discussion
- Is there any record of spontaneous hybridization between Kalanchoe species in Ecuador? What is the origin of the hybrids recorded in Ecuador (artificial from cultivation?). Please, discuss these issues.
Materials and methods
- page 15, line 539: Please, explain that the name Bryophyllum is a synonym of Kalanchoe. What about the other synonyms of the genus and lower taxa?
- page 16, line 580: Please, change the name of subchapter as “Categorization of Kalanchoe taxa by alien status”
- page 16, lines 587-589: it is important to indicate that the alien species should form self-replacing populations without direct human intervention for a period of at least 10 years to be classified as naturalized
Some style and editing corrections are needed:
- in Abstract: delete the bold style in “The genus”
- the word “wildlife” should be replaced by “wild” and carefully checked throughout the whole text, including the title, captions of figures and tables, as well as the content of graphs and tables, and description of supplementary materials
- please, choose one version of the name of the Galapagos Islands (there are other versions like Galápagos Islands, the Galapagos).
- “neatly invasive” should be corrected as “invasive”
- page 16, lines 549-550: correct “OR” and “AND” with small letters
- page 17, line 598: add (HF) after “human footprint’
- page 17, line 629: correct “principal components analysis” as “principal component analysis”
- Table 1 as a whole should be placed on one page
- Supplemental figures: correct “Figura” as “Figure” [the figures are hard to read, when zoomed in, they blur a bit]
- Supplemental tables: replace “species” by “taxa” since the hybrids are also included
